# Mechano-adaptive meta-gels through synergistic chemical and physical information-processing

**Brigitta Dúzs** ✉, **Oliver Skarsetz**, **Giorgio Fusi, Claudius Lupfer &**
**Andreas Walther** ✉

Global functional adaptation after local mechanical stimulation, as in mechanobiology and the mimosa plant, is fascinating and ubiquitous in nature. This is achieved by locally sensing mechanical deformation with precise thresholds, processing this information via biochemical circuits, followed by downstream actuation. The integration of such embodied intelligence allowing for mechano-to-chemo-to-function information-processing remains elusive in man-made systems. By merging the fields of chemical circuits and metamaterials, we introduce adaptive metamaterial hydrogels (meta-gels) that can accurately sense mechanical stimuli (local touch and global strain), transmit this information over long distances via reaction-diffusion signaling, and induce downstream mechanical strengthening by growing nanofibril networks, or soft robotic actuation through competitive swelling. All elements of the sensor-processor-actuator system are embedded in the device, functioning autonomously without external feeding reservoirs. Our concept enables designing advanced life-like materials systems that synergistically combine two worlds – chemical circuits for chemical information-processing and metamaterial unit cells for physical information-processing.

Living entities interact with and adapt to their environment by applying the principles of decentralized embodied intelligence using a sensor-processor-actuator paradigm[1,2]. This paradigm inspires the next generation of life-like systems with unprecedented capacity for adaptation and autonomous operation. For example, sensing of mechanical forces allows to detect obstacles for guiding movement or to change physical properties[3,4]. The underlying information-processing, i.e., mechanotransduction, is a multi-step mechanism in which force events are converted into (bio)chemical signals and processed in nonlinear biochemical signaling circuits to finally achieve mechanical adaptation as an actuator function[5–7]. The mimosa plant is an excellent example of local sensing, long-range information-transmission, and global adaptation through actuation[8]. In stark contrast, classical responsive materials do not allow for equally complex behavior because they lack processor functions. The progress in materials

relevant to this study has focused so far mainly only on the mechano-sensing and actuation parts, such as mechanoresponsive polymers that exhibit continuous activation of bonds without a critical threshold, and soft robotic devices that can bend under a direct external stimulus[9–16].

The challenge of incorporating information-processing elements into mechanical materials can be tackled from different fields. Complex autonomous decision-making requires nonlinearities in chemical or physical dynamics, as in chemical reaction networks (CRNs) or mechanical metamaterials[1,2,17–20]. CRNs use chemical feedback loops to generate exotic behavior, e.g., self-acceleration through positive feedback or transient states and oscillations through negative feedback[21–23]. CRNs combined with transport processes result in reaction-diffusion (RD) phenomena such as propagating chemical signals or self-organized patterns[17,24]. CRNs have been used to program autonomous

Life-Like Materials and Systems, University of Mainz, Duesbergweg 10-14, Mainz 55128, Germany. ✉e-mail: brigitta.duzs@uni-mainz.de;
andreas.walther@uni-mainz.de

lifetimes and periodic behavior in self-assemblies and materials[25–29]. However, in these cases, typically, a homogeneous chemical signal dictates a spatially uniform bulk material behavior. The development of materials with a fully integrated mechanotransduction system that enables precise sensing of a macroscopic deformation state and local-to-global adaptation remains an unresolved challenge, even though it is highly relevant technologically. Mechanical metamaterials offer a complementary perspective on the topic. Metamaterials use porous unit cell structures with specific geometries to produce nonlinear mechanical response impossible for bulk materials[18,30–33]. These non-linearities can be used for logic gates, signal propagation, and processing. This area is largely driven by solid mechanics[34]. Combining it with the world of CRNs and responsive materials can open unprecedented possibilities for synergistic chemical and material intelligence in next-generation life-like material systems.

Herein, we introduce a platform concept to realize the sensor-processor-actuator paradigm in soft robots and mechanical materials, integrating the RD dynamics of a pH-autocatalytic CRN[23] as the information-processing element with metamaterial unit cells as sensory elements. Autocatalysis serves as a signal enhancement, and the coupling to diffusive transport allows for a sharp, self-sustaining chemical front, which ensures local-to-global signal transmission throughout the material[35,36]. The sensing step is the local detection of a mechanical force. We design a metamaterial unit for force-threshold-dependent activation. The actuation, that is, mechanical strengthening and shape-morphing, is achieved by using pH-triggered materials that react downstream of the CRN processor. This concept allows us to incorporate nontrivial spatiotemporal information-processing into materials and build autonomous, freestanding soft robots with mimosa-like system-level adaptivity induced by local forces.

## Results

### Integrated system design

Our meta-gel systems integrate three distinct material and information-processing concepts for autonomous operation and downstream functional adaptation: (1) metamaterial structure as strain gates, (2) CRN for RD signal transmission, and (3) responsive muscle elements (Fig. 1). The general operational principles focus on mechano-chemo-mechano communication and information-transmission from local events to global behavior: Local sensing happens when the target hydrogel device, as illustrated by a robot arm, is acted upon by an external force (red dot). In contrast to simple external touching, this sensor event is embedded within the device using a structural meta-material element that allows for installing a strain/stress threshold for contact. During local contact, the force is translated to a chemical

signal because the activating chemical species (OH⁻) diffuses from the touching patch to the target device (Fig. 1a). However, since diffusive transport in cm-size devices is inefficient and quickly decays with time and distance, it is necessary to amplify the signal. Therefore, we incorporate an enzymatic urea-urease CRN into the meta-gel to auto-catalytically amplify the OH⁻ signal by conversion of urea into $NH_3$ and $CO_2$[35,37]. Urea is the latent chemical energy source driving the long-range information-transmission. Critically, the CRN remains in a dormant state in the acidic starting regime in the absence of initiation, but a sharp high-speed RD front with a constantly self-amplifying OH⁻ signal propagates with spatiotemporal control over the device when touched with a basic patch (Fig. 1b, CRN in Fig. 2a). Finally, pH-induced downstream processes translate back the chemical information into a structural and mechanical response as adaptation. We showcase the shape morphing of a soft robot or mechanical self-strengthening due to nanofiber formation (Fig. 1c). In summary, the overall system first translates mechanical and structural information from a local spot into a fuel-driven spatiotemporal chemical amplifier, then it finally translates the chemical information on a global scale back into structural and mechanical adaptation.

### Autonomous actuators with soft touching and local-to-global transmission using RD fronts

Figure 2a, b detail the urea-urease CRN that autocatalytically amplifies OH⁻ from urea[35,37]. Urea-urease is dormant at low pH due to the pH-dependent activity profile of urease. When urea-urease is embedded in a non-convective environment (e.g., hydrogels) at its dormant pH state, the addition of small amounts of base triggers rapid auto-catalytic OH⁻ production, which propagates as an RD front transmitting a basic pH signal in space and time. We integrated the urea-urease CRN into multi-material hydrogel devices using custom-made molds and sequential polymerization (Methods and Figs. S1, S2). The hydrogel domain responsible for the RD front was obtained by crosslinking star-shaped polyethylene glycols (sPEG, see Supplementary Notes 1) in the presence of urea, urease, and a pH-stabilizing citrate buffer of pH = 2.75. The low pH maintains the urease in its inactive window where it cannot process urea so it preserves the initial dormant chemical state of the entire device. Bromocresol purple (BCP; pH indicator) visualizes the RD front propagation with a pH flip from 3 to 9 (yellow: initial acidic; purple: reacted basic state; concentrations are in Table 1). Figure 2c, d depict a propagating pH RD front in a gel stripe after initiation by a drop of 0.1 M NaOH solution on the left. The kymograph gives a constant speed of 0.11 mm min⁻¹ (Fig. 2d), which in principle can be tuned by the urease concentration[38,39]. This sets the basis for the long-distance autocatalytic information-transmission.

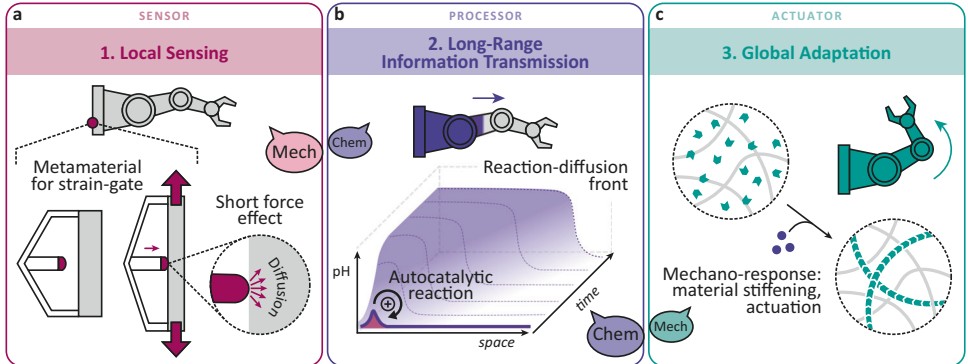

**Fig. 1 | Chemo-mechanical and local-to-global information-processing based on the sensor-processor-actuator paradigm for the design of adaptive hydrogel muscles. a** Local mechanical trigger via soft touching and short-range diffusive transport. Metamaterial design for the strain-gated mechano-activation. **b** Chemical signal amplification and spatiotemporal transmission using an autocatalytic reaction-diffusion front. **c** System-level mechanical reprogramming and structural adaptation due to a downstream actuation. The formation of supramolecular polymers is one example. The speech balloons illustrate the translation of mechanical input to a chemical transportable signal that is finally translated back to a mechanical output.

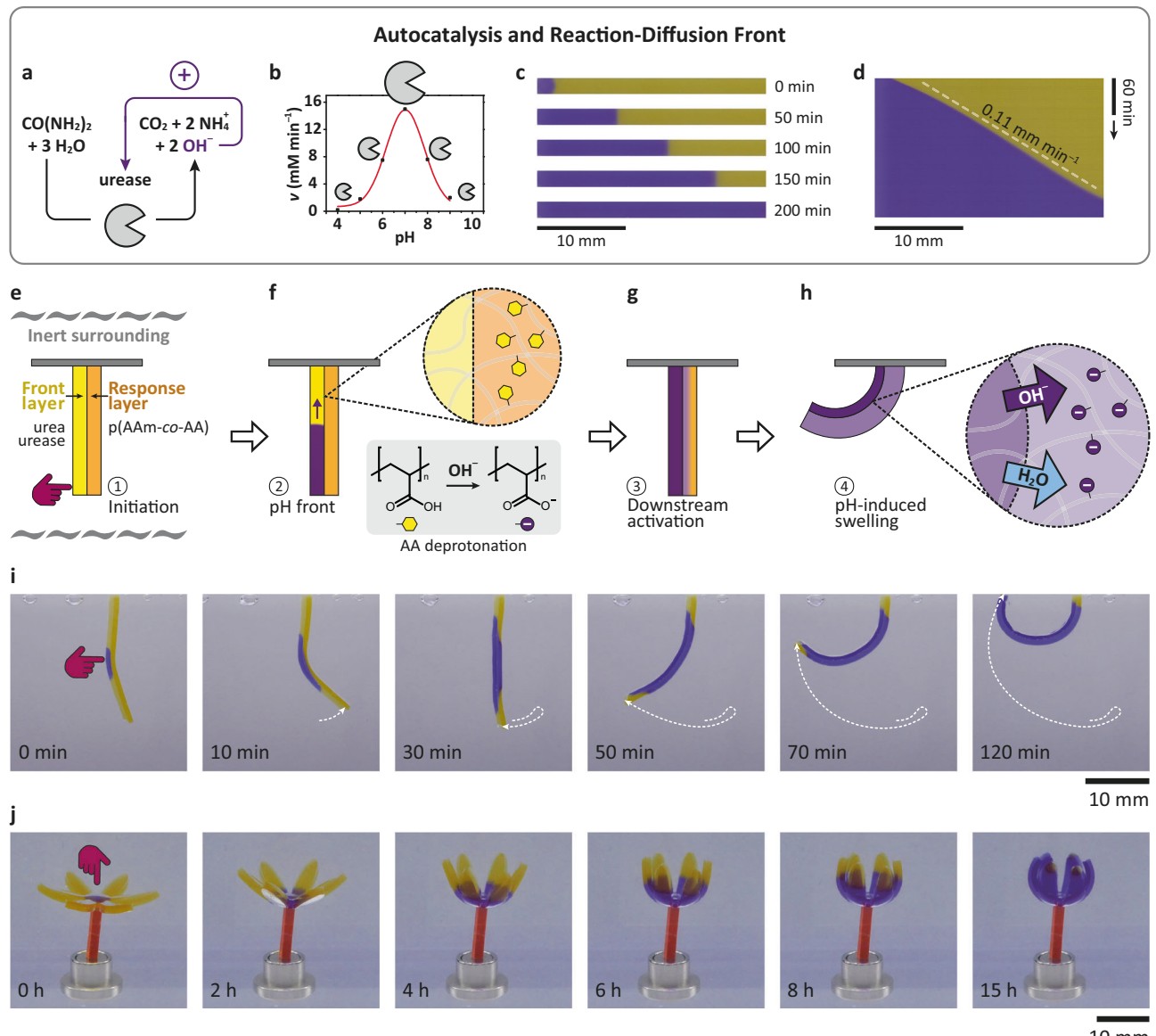

**Fig. 2 | Reaction-diffusion (RD) front to drive bilayer actuators. a–d** Signal amplification and spatiotemporal transmission using an autocatalytic RD front: **a** Scheme of the urea-urease autocatalytic chemical reaction network, **b** pH-dependent activity ($v$) of the urease enzyme, **c** timelapse snapshots and **d** kymograph of RD front propagation in a hydrogel stripe. **e–h** Scheme of the operation steps: **e** Initial dormant state before the soft touching. The bilayer consists of the front layer with urea and urease, and the response layer with pH-responsive p(AAm-co-AA) and is immersed in inert silicone oil (gray waves, removed for clarity in the next images). **f** OH⁻-autocatalytic front propagation in the front layer. **g** OH⁻ diffusion from the front layer to the connected response layer. **h** Bending due to the swelling of the response layer: water uptake from the connected front layer. **i** Experimental timelapse of the touching-induced bilayer actuation. **j** Self-protective mechanism in a bilayer flower (bottom: response, top: front layer). Each layer is 0.75 mm thick at the beginning. Curve in (**b**) was adapted from our previous work[37]. Details of compositions are in Methods and Table 1. AAm acrylamide, AA acrylic acid.

Next, to demonstrate the integration into higher-level functional devices, we designed touch-activated autonomous bilayer actuators as a simple sensor-processor-actuator material system by combining sPEG-urea-urease pH front layers with pH-sensitive response layers (Fig. 2e–j). The response layer was photopolymerized from a mixture of acrylic acid (AA, pH-sensitive), acrylamide (AAm, passive), and crosslinker (Methods, Supplementary Notes 2). The compositions of the front and response layers were adjusted for similar swelling to ensure the initial state of the bilayer actuator to be vertically straight. We used an inert oil to avoid drying during the experiments and to demonstrate the autonomous nature. Critically, shape changes during actuation are dictated by water exchange between the layers, and not by exchange to a surrounding bath. Hence, our devices differ profoundly from usual hydrogel bilayer actuators[40] in that they (1) operate fully autonomously, without any external feed from the surrounding liquid and (2) none of the layers is a passive element, both have a distinct function in the operation of the bilayer.

Figure 2e–h show the working principle of such a device. First, the bilayer is straight with a metastable, dormant state of the RD system. Then initiation happens by touching with a high-pH hydrogel piece. During the mechanical contact (Fig. 2e), a catalytic amount of OH⁻ diffuses into the front layer and activates the front propagation, leading to autocatalytic OH⁻ amplification (Fig. 2f). The amplified [OH⁻] is sufficient to basify the response layer by short-range diffusion (Fig. 2g). This triggers the downstream processes, which start on a chemical level via deprotonation of the AA moieties (p$K_a \approx 4.25$) and transformation of the pH-sensitive response layer into a polyelectrolyte gel. This is followed on a material level by rapid osmotic

**Table 1 | Initial concentrations of the embedded chemical reaction network species in experiments**

| Experiment number and device parts | | [urea] (mM) | [urease] (g L⁻¹) | [HCl] (mM) | [NaOH] (mM) | [Fmoc-EDA] (g L⁻¹) | [ThT] (mM) |
|---|---|---|---|---|---|---|---|
| Fig. 2c, d | Gel stripe | 35 | 0.62 | 4.00 | – | – | – |
| Fig. 2i | Front layer | 55 | 1.42 | 4.00 | – | – | – |
| | Response layer | 55 | – | 1.25 | – | – | – |
| Fig. 2j | Front layer | 55 | 1.20 | 4.20 | – | – | – |
| | Response layer | 55 | – | 1.25 | – | – | – |
| Fig. S4 | Front layer | 55 | – | 4.00 | – | – | – |
| | Response layer | 55 | – | 1.25 | – | – | – |
| Fig. 3i | Front layer | 55 | 1.34 | 4.00 | – | – | – |
| | Response layer | 55 | – | 1.25 | – | – | – |
| | Strain-gate basis | 55 | – | – | 2.40 | – | – |
| Movie 3 | Front layer | 55 | 1.60 | 4.10 | – | – | – |
| | Response layer, stem | 55 | – | – | – | – | – |
| | Response layer, leaves | 55 | – | – | – | – | – |
| Fig. 4b–d | Strain-gate basis | 55 | 0.92 | 4.00 | – | 10 | – |
| Fig. 4e | Strain-gate basis | 55 | 0.92 | 4.00 | – | 10 | 1.0 |

*Fmoc-EDA* Fmoc-ethylenediamine hydrochloride, *ThT* thioflavin T.

swelling of the response layer that takes up water from the RD front layer. Concomitantly, the RD front layer shrinks and the bilayer bends as a mechanical downstream process (Fig. 2h). In real experiments the front propagation and the basification of the response layer occur in parallel. Such a design is rare but can be achieved using other chemistries as well. An autocatalytic front-mediated hydrogel bilayer has been constructed using thiol fronts in the front layer and reduction of disulfide crosslinks in the response layer. There the actuation mechanism is different, based on the release of the mechanical strain of the initial bilayer state[36].

Experimentally we demonstrate a bending arm and a self-protecting flower (Fig. 2i, j). The bilayer arm was poked at a central position (but any position is possible) for 10 sec with a 1 mm *N*-[2-(dimethylamino)ethyl]acrylamide (DMAEAAm) hydrogel (pH = 10). The triggered purple RD front propagation is visible in the front layer (left side, Fig. 2i and Supplementary Movie 1). In the first 10 min, the bilayer slightly bends to the right due to the generation of osmotically active products (urea is converted into $(NH_4)_2CO_3$ and $NH_4OH$; $CO_2$ may evaporate during the process), leading to osmotic swelling of the RD front layer. The expected bending to the left starts after 10 min, when the response layer (right side) turns purple due to diffusion of the basic urea-urease products across the full bilayer. The diffusion of $OH^-$ tips the balance for osmotic swelling as the pH-sensitive response layer becomes ionized, and the resulting polyelectrolyte swelling induces strong bending to the left. This behavior is the consequence of the water exchange between the layers (and not with the environment). The thickness of the response layer increases by 22% while that of the front layer decreases by 22% during the actuation (compare Fig. 2i, 0 min and 120 min). The actuation is larger for a higher [AA]:[AAm] ratio (Fig. S3). [AA] must not exceed certain thresholds, as the AA can lead to a buffering effect detrimental to the autocatalytic front propagation. An increase in [urea] and [urease] can help to balance these effects, and we typically constructed robust devices at [AA] = 100 mM in the response layer. Without the urease enzyme, the processor mechanism is missing, and the bilayer does not actuate (Fig. S4).

In the self-protecting flower design, we prepared flower petals from a 2D bilayer in horizontal orientation with RD front layer on top and response layer below (Fig. 2j and Supplementary Movie 2). The self-closing mechanism is triggered by a local and quick touch signal in the flower center. The RD front propagates in a quasi-2D plane, eventually causing concentric bending along the petals. We used lower initial [urease] in the RD front layer to achieve a slower RD signal

processing for this device (Table 1). All petals close synchronously, indicating a high reproducibility of the RD front propagation and actuation. Here the adaptation mechanism hides and protects the area where the signal was sensed. This example demonstrates the versatility of our robot concept in varying 1D and 2D geometries. Supplementary Movie 3 shows a more complex multi-material design mimicking the sequentially closing leaves of the mimosa plant. With other customized device shapes as known from classical soft robotics, it is possible to achieve further versatile autonomous task operations (e.g., lifting, secondary touching, wrapping)[41,42].

## Metamaterial threshold strain gate as material-embedded sensor

Next, we turn to the fundamental challenge of installing a precise strain gate for better control over the mechano-to-chemo activation interface. In the field of mechano-adaptive polymer materials, mechano-activated bond rupture has given rise to mechanochromic behavior or other mechano-triggered processes[9–16,28,43]. Even the mechano-activated release of basic compounds has been described, which might appear as an interesting starting point for triggering basic pH fronts[44]. However, such classical bulk scale mechano-triggered processes occur across all strain states when stretching materials and are thus prohibitive of installing precise strain gates with a clear all-or-nothing threshold. Therefore, we took inspiration from metamaterial concepts, where unit cell mesostructures define functional properties as a form of physical intelligence[2,32,33]. In more detail, we designed small metamaterial sensor units with an integrated high pH patch that can only activate the pH RD system after reaching a critical strain threshold (Fig. 3a–e). This topological strain gate is an analogy to mechanical metamaterials with positive Poisson's ratio, where the tensile strain in one direction produces compressive strain along the perpendicular axis[31].

We first modeled the strain gate behavior using finite element simulations (FES) and coupled the mechanical part of the FES with the RD front propagation by implementing a simple autocatalytic reaction into the body (Fig. 3a–g and Supplementary Movie 4). In the initial resting state, there is a small gap (no touching) between the high-pH patch (purple) and the main low-pH meta-gel body (yellow). When stretched horizontally, the gap size decreases vertically, and eventually, the two parts contact. This is a sharp threshold for the initiation because the activation happens if, and only if, the diffusive contact is established. The arising sharp purple RD front is the consequence of the autocatalytic self-amplification of the transferred signal ($OH^-$). The

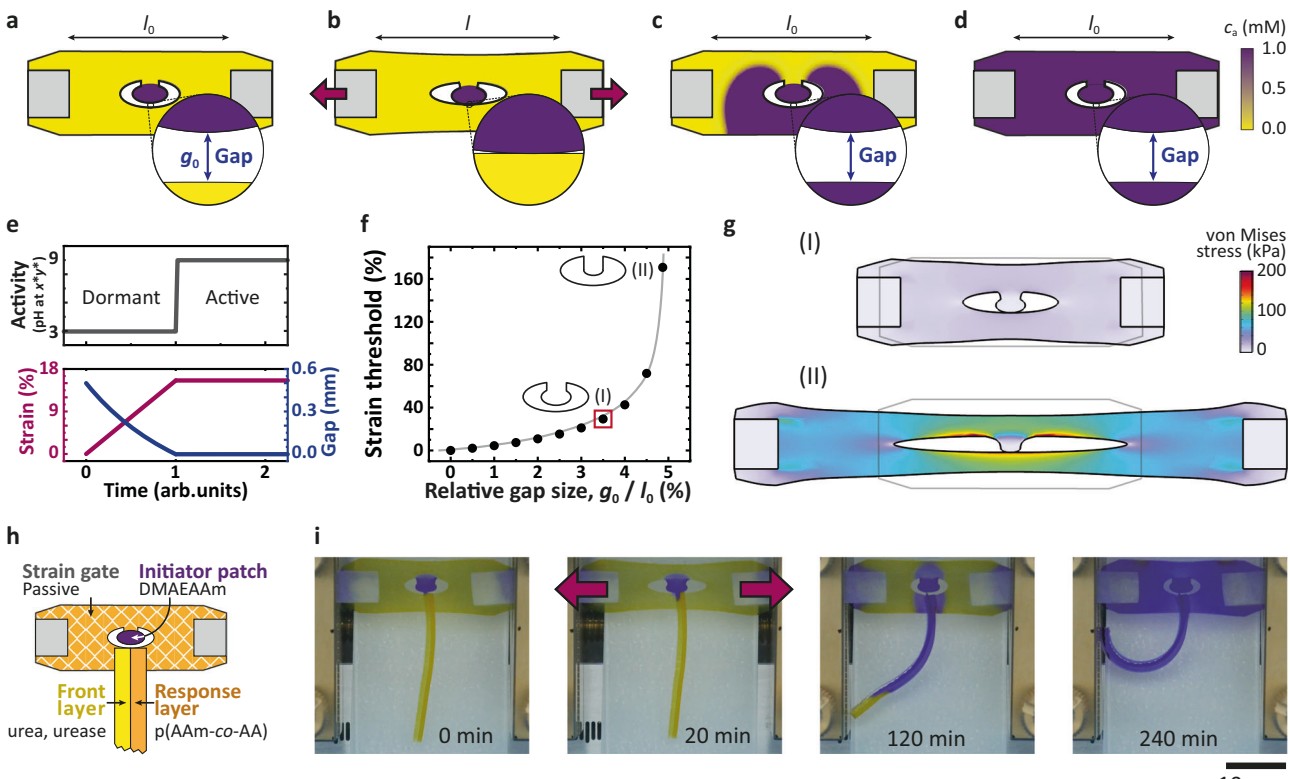

**Fig. 3 | Remote activation of a soft robotic bilayer using distant metamaterial strain gates and chemical information-processing. a–g** Geometry optimization by finite element simulation: **a–d** Step-by-step operation principle of the strain gate with an autocatalytic front. Purple indicates high activator concentration ($c_a$). **e** The gradual increase of strain and decrease of gap size results in a sharp activation threshold. **f, g** The strain-threshold needed for activation exponentially increases with the initial relative gap size ($g_0/l_0$). The geometry corresponding to the experiments is highlighted by a red square. The images show the initial (gray frame) and final (colored) states of the stretching. **h** Scheme of the combined experimental system of the strain gate and the bilayer actuator. **i** Timelapse of the strain-gated soft robot bending with a remote metamaterial sensor (initial gap size, $g_0 = 0.5$ mm, initial distance between clamped points, $l_0 = 15$ mm). Details of compositions are in Methods and Table 1. AAm acrylamide, AA acrylic acid, DMAEAAm N-[2-(dimethy-lamino)ethyl]acrylamide.

RD front efficiently transmits the local information all over the spatial domain, even if the original patch contact is removed (Fig. 3c, d). Figure S5 and Supplementary Movie 4 show a comparison to a purely diffusive front where information transfer breaks down. More importantly, FES reveals universal scaling laws between the strain threshold for touching and the width between the high-pH patch and the rest of the body ($g_0$) relative to the length of the specimen ($l_0$). The strain threshold for activation exponentially increases with the initial relative gap size (Fig. 3f, g).

For the experiments, we chose 0.5 mm as a suitable gap width for a device with a body of 8.8 mm × 17.6 mm. Such a gap is large enough to avoid unwanted touching during device assembly and small enough to avoid excessive stress concentration and potential fracture during stretching of the meta-gel. We implemented the metamaterial strain gate directly together with a bilayer to demonstrate a strain-gated remote initiation of a soft robotic actuator (Fig. 3h, i). Its high-pH patch was made from crosslinked p(AAm-co-DMAEAAm). The DMAEAAm sets the patch pH to 10, where the OH⁻ ions are counterions that diffuse slowly (Supplementary Notes 3). This ensures that the activation of the connected bilayer happens only due to the strain-gated soft touching. The rest of the strain gate was made from crosslinked pAAm, buffered at pH = 2.75 with citrate buffer, and containing BCP indicator, urea, but no urease. The urea-urease bilayer was connected to the strain gate basis over a 3 mm section (Fig. 3h). To trigger the bilayer device, we stretched the module to the touching point. After successful initiation, the touching was removed while the self-sustaining RD front propagated along the bilayer, followed by downstream actuation (Fig. 3i). The snapshots reveal that the actuator

bends locally as the front propagates along till full propagation and bending. Over time, small amounts of OH⁻ leak from the DMAEAAm patch and from the amplified basic front in the bilayer into the passive body with the strain gate. Overall, in striking contrast to classical chemically triggered bilayer actuators that require global exposure to a stimulus[14,40], we herein present the first remote mechanical activation scenarios of chemically operated bilayer soft robots via long-distance signal transmission and even following a strict-logic strain gate response.

## Strain-gated spatiotemporal hydrogel stiffening

Next, we demonstrate strain-gated adaptation towards mechanical self-reinforcement by downstream self-assembly of stiff nanofibers in the main body of a meta-gel. Due to the previously detected diffusive leakage into the body of the strain gate (Fig. 3i), we now isolated the high-pH activation patch from the body with a p(AAm-co-AA) neck buffered to pH = 2.75. The rest of the body was made from a sPEG-hydrogel loaded with the urea-urease autocatalytic pH RD front system. Additionally, in the main body, we incorporated Fmoc-ethylenediamine hydrochloride (Fmoc-EDA), which is known to assemble into nanofibers above pH 8.4 due to charge removal at the amine function[45]. In contrast to the distant sensing of Fig. 3, here, the strain gate for autocatalytic [OH⁻] front initiation is located within the hydrogel body that we want to strengthen (Fig. 4a, b). FES illustrates the process (Fig. 3a–d). The experiments proceed accordingly: The RD front is initiated after reaching the predicted threshold during tension (Fig. 4b). As the OH⁻ signal propagates through the material, deprotonation and self-assembly of Fmoc-EDA into nanofibers occurs

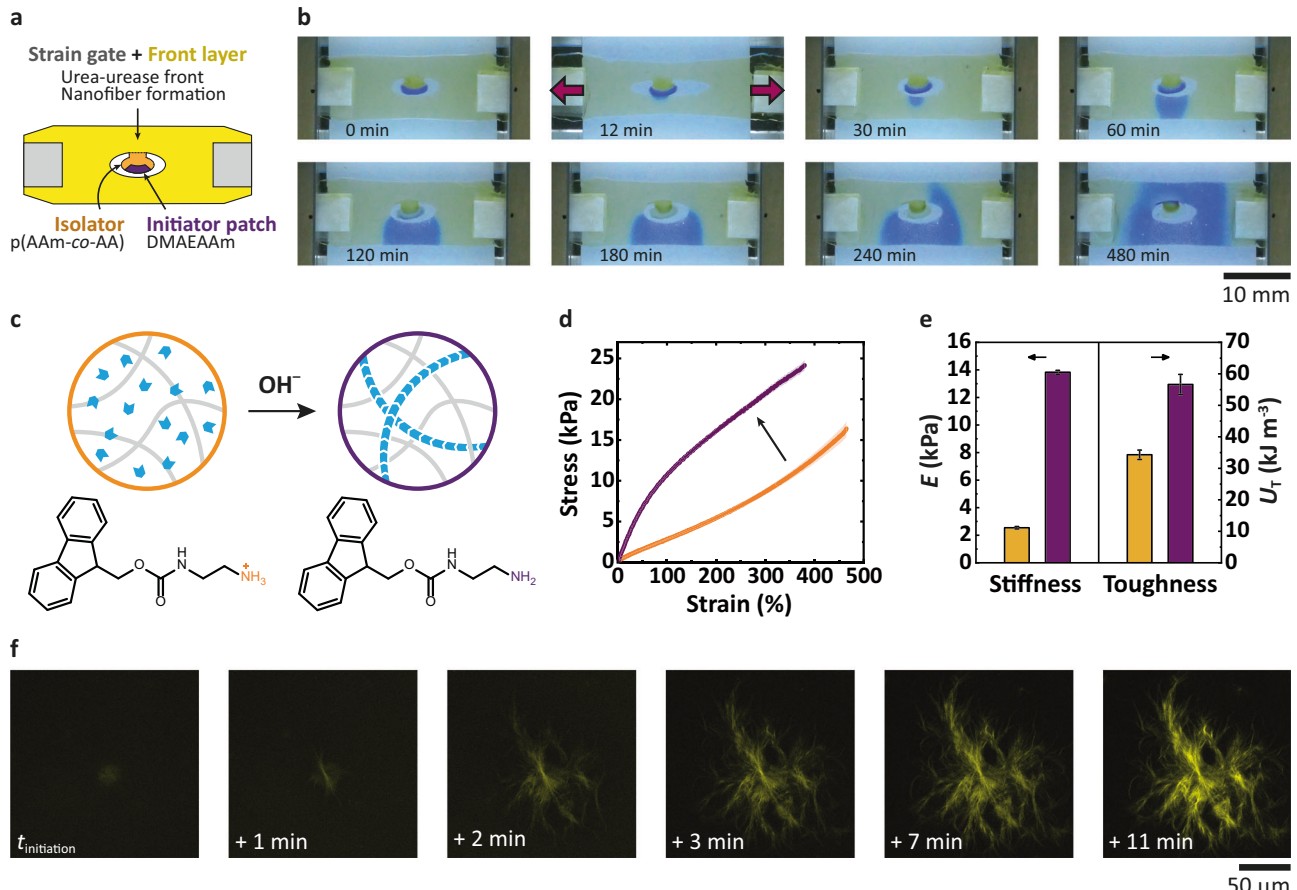

**Fig. 4 | Strain-gated global mechano-adaptation due to nanofiber self-reinforcement after local touching event. a** Scheme of the device with an isolated strain gate. **b** Timelapse snapshots of the OH⁻ front propagation in the strain-gated meta-gel. **c** OH⁻-induced self-assembly of nanofibrils inside the hydrogel. **d**, **e** Mechanical properties of the hydrogel before (yellow) and after (purple) the OH⁻ front. **d** Tensile tests. **e** Stiffness ($E$) and toughness ($U_T$) changes during adaptation. The error bars represent standard error. **f** Nanofiber growth in the hydrogel monitored by confocal laser scanning microscopy (Thioflavin T staining), where $t_{initiation}$ is the time of basification in the monitored volume of the gel. Details of compositions are in Methods and Table 1. AAm acrylamide, AA acrylic acid, DMAEAAm *N*-[2-(dimethylamino)ethyl]acrylamide.

as a downstream regulatory actuator to reinforce the mechanical properties of the whole metamaterial based on the local touching event (Fig. 4c).

The locally induced but globally occurring mechano-adaptation can be quantified by comparing the tensile properties of the initial and the final states (Fig. 4d, e). The Young's modulus ($E$, stiffness) increases sevenfold, and the toughness ($U_T$, deformation energy) doubles due to the nanofiber formation, confirming an excellent strengthening effect. In situ rheology in Fig. S6 further demonstrates that fiber reinforcement occurs similarly quickly as the RD front propagation. Confocal laser scanning microscopy (CLSM) confirms the quick fiber assembly within 1 min after the pH flip and completed structures within ca. 7 min (Fig. 4f). The further increase in fluorescence intensity rather stems from additional Thioflavin T (ThT) incorporation used to stain the fibers.

In the context of mechano-adaptive materials with self-strengthening behavior, it is important to realize that the FES-assisted metamaterial design presented here not only results in materials that undergo reinforcement when a specific strain is reached. However, since strain and stress are intrinsically correlated parameters for materials, this also means that a critical tensile force is actually used as a threshold to sense when the mechanical strengthening effect should occur. This brings us one step closer to mimicking the non-trivial, gated force-dependent decision-making of living systems, and also provides a pathway for self-protection in technical applications.

## Discussion

The implementation of the sensor-processor-actuator paradigm with autonomous information-processing and operation inspired by the embodied intelligence of living systems is one of the greatest challenges in engineering new functional soft materials. New generic platform concepts are needed, where the complex material-level decision making (thresholds, response amplitude, pathways, etc.) arise from relatively simple nonlinear elements. Such elements of embodied intelligence can be provided with different approaches, e.g., CRNs, metamaterials, or also neuromorphic semiconductor devices, where each domain has different strengths[1,2]. In this work, we merged elements of two otherwise separated fields of embodied intelligence—autocatalytic CRNs and metamaterial strain gates—to realize a synergistic combination of nonlinear modules for the design of adaptive materials systems. Unlike previous studies focusing on bulk programmability[25–29], our approach realizes spatiotemporal sensing, information propagation, and structural actuation in spatially mesostructured meta-gels. This is crucially important for advanced local-to-global information-transmission and adaptation scenarios.

The information-transmission system in this work is empowered by the urea-urease OH⁻-autocatalytic front reaction. Autocatalysis multiplies the signal chemically, and the coupling with diffusive transport results in sharp, long-distant RD fronts. The RD mechanism ensures fast and non-exhausting signal transmission from a local sensory event to all over the object, regardless of its size. In addition, the

enhanced chemical signal is a strong enough effector to trigger a downstream process without killing the front. This strategy may be expanded to other autocatalytic processes as long as the initial dormant state is sufficiently stable. However, the advantage of the urea-urease system is its biocompatibility and the availability of many pH-responsive systems to engineer downstream functional processes. Here, we demonstrated applications in soft robotics using pH-responsive polymers and in self-strengthening materials exploiting pH-triggered nanofibrillation.

Nontrivial life-like behaviors always arise from non-equilibrium operation. The actuation speed is limited by the front propagation that is engineered to be slow enough to provide a sufficiently long-lasting dormant state, and the water transport responsible for swelling. Therefore, CRN-empowered soft robots have a slower response and smaller generated forces than hard robots. However, they are especially suitable for creating complex autonomous (computer-chip-free) dynamics that mimic living systems, and the generated forces are potentially enough to interface them with biosystems on a small size scale. Our gel devices presented here respond to one-time events. Resettability and cyclic operation would be extremely challenging in such a closed, compartmentalized system. The current state of pH CRNs does not provide suitable activator-inhibitor-type chemical dynamics for closed systems; a potential solution could be external refueling using vasculature or further hydrogel elements.

From a wider perspective, mechanical stimulation is one of the most ubiquitous sensory inputs in fields from biomaterials to macroscopic structural materials, but building the interface from mechanics to CRNs and back to engineer life-like functions remains largely elusive. The hydrogel framework is excellent for confined, freestanding 3D devices that can naturally process force and touching events and robustly accommodate spatiotemporal chemistry. In this work, we first created hydrogel actuators sensing and processing local non-thresholded compressive forces, and secondly, we transformed stretching into thresholded compression with the strain gate. We introduced how comparably simple metamaterial unit cells can generate fundamentally new self-controlled behavior allowing for (1) distinct thresholding of activation strain and (2) also for delocalizing sensor units away from the actor unit. Distant off-robot activation opens fundamentally new design opportunities in soft robotics, where sensing, processing, and acting are spatially separated and thus can be coupled to different input and output signals (e.g., mechanical and chemical). Depending on the strength of the mechanical signal (strain and stress), our system reacts in a strictly binary way: In the case of low strain, it remains unchanged, but in the case of high strain, it adapts everywhere. We leveraged this behavior for self-strengthening materials that become stiffer and tougher once they are stretched to a supercritical level. We believe that the great progress in mechanical metamaterials and 3D printing opens possibilities for more complex material designs. Looking out to the future: The presented concept, integrating RD signaling, metamaterial sensing, and adaptive downstream processes, offers a new perspective for future multi-sensory soft material constructions, enabling quasi-intelligent fate-selection and complex autonomous decision-making and self-controlled operation in space and time.

## Methods
### Materials
Acrylamide (AAm, 99%, Sigma-Aldrich), *N,N*'-methylenebisacrylamide (Bis, 97%, Alfa Aesar), *N*-[2-(dimethylamino)ethyl]acrylamide (DMAEAAm, 98%, TCI), acrylic acid (AA, 99.5%, Acros Organics), ethyl (2,4,6-trimethylbenzoyl) phenylphosphinate (95%, Biosynth), lithium bromide (99%, Alfa Aesar), poly(ethylene glycol) (6 kg mol$^{-1}$, PEG6k, for synthesis, Sigma-Aldrich), 4-dimethylaminopyridine (99%, Alfa Aesar), *N,N*'-dicyclohexylcarbodiimide (99%, Aldrich), 4-(11,12-didehydrodibenzo[*b,f*]azocin-5(6H)-yl)−4-oxobutanoic acid (DBCO-COOH) synthesized according to a previously reported procedure[46], 4-arm-

NH$_2$HCl-PEG20k (20 kg mol$^{-1}$, JenKem Technology), hexafluorophosphate azabenzotriazole tetramethyl uronium (HATU, 98%, Carl Roth), 4-methyl-morpholin (99%, Sigma-Aldrich), 4-arm-OH-PEG20k (20 kg mol$^{-1}$, JenKem Technology), triethylamine (99.5%, Sigma-Aldrich), methanesulfonyl chloride (MsCl, 99.7%, Sigma-Aldrich), sodium azide (99.5%, Sigma-Aldrich), *N,N*-dimethylformamide (DMF, 99.8%, extra dry over molecular sieve, Thermo Scientific), dichloromethane (DCM, 99.8%, extra dry over molecular sieve, Thermo Scientific), urea (99%, Sigma-Aldrich), urease (from *Canavalia ensiformis*, type III, 15,000–50,000 units g$^{-1}$, Sigma-Aldrich), Fmoc-ethylenediamine hydrochloride (Fmoc-EDA, 95%, abcr), Thioflavin T (ThT, Sigma-Aldrich), bovine serum albumin (BSA, Sigma-Aldrich), bromocresol purple (BCP, 90%, Sigma-Aldrich), citric acid (99%, Acros Organics), trisodium citrate dihydrate (99%, Acros Organics), NaOH (98%, Sigma-Aldrich), HCl (1 M, VWR), silicone oil (medium viscous, 350 cSt, Carl Roth). The solvent was MilliQ water if not indicated otherwise. The BCP stock solution was made by using 10 vol% ethanol and 90 vol% water. The purchased chemicals were used without further purification. The following chemicals were synthesized based on published procedures: lithium phenyl-2,4,6-trimethylbenzoylphosphinat (LAP), poly(ethylene glycol) diacrylate (PEGDA6k), 4-arm-DBCO-PEG20k, and 4-arm-N$_3$-PEG20k.

### Synthesis procedures
**Synthesis procedure of LAP.** The synthesis was carried out according to a previously reported procedure[47]. Briefly, ethyl (2,4,6-trimethylbenzoyl) phenylphosphinate (5.69 g, 18.0 mmol) was dissolved in 2-butanone (100 mL). Lithium bromide (6.25 g, 72.0 mmol) was added, and the mixture heated to 50 °C for 10 min, allowed to cool and left to stand overnight. The crystallized product was recovered by filtration, washed with ice-cold 2-butanone, and dried in vacuo to yield LAP as a fine white powder in quantitative yield.

**Synthesis procedure of PEGDA6k.** PEG6k (50.0 g, 8.30 mmol) was dried by heating to 80 °C in vacuum overnight. To the dried and cooled PEG6k, dry DCM (200 mL), 4-dimethylaminopyridine (0.300 g, 2.50 mmol), and acrylic acid (3.00 g, 3.40 mL, 41.6 mmol) were added. The mixture was allowed to homogenize and cooled to 0 °C. Dicyclohexylcarbodiimide (12.8 g, 62.5 mmol) was added, and the mixture stirred for 4 h at 0 °C, and at room temperature for 48 h. The precipitated dicyclohexylurea was removed by filtration, the volume of the solvent was reduced in vacuo and the crude product precipitated out of diethyl ether. The crude product was redissolved in acetonitrile, filtered twice to remove additionally precipitated dicyclohexylurea, and the product reprecipitated from diethyl ether. The product was recovered as a white powder (42.0 g, 6.90 mmol, 84% yield).

**Synthesis procedure of 4-arm-DBCO-PEG20k.** The synthesis was carried out based on a previously reported procedure[48]. DBCO-COOH (61.1 mg, 0.200 mmol, 8.00 eq.) and HATU (111 mg, 0.293 mmol, 11.7 eq.) were dissolved in 1 mL anhydrous DMF under N$_2$. 4-methylmorpholin (75.0 μL, 0.675 mmol, 27.0 eq.) was added to this solution and reacted for 15 min at room temperature under N$_2$. In parallel, 4-arm-NH$_2$HCl-PEG20k (500 mg, 0.025 mmol, 1.00 eq.) was dissolved in 3 mL anhydrous DMF under N$_2$. Then the activated DBCO-COOH/HATU solution was added dropwise to the 4-arm-NH$_2$HCl-PEG20k solution. The resulting mixture was stirred overnight at room temperature under N$_2$. The reaction mixture was concentrated under reduced pressure, dissolved in MilliQ water, dialyzed against water for 48 h (ZelluTrans, MWCO = 6–8 kDa, wall thickness 30 μm), filtered, and lyophilized. The product was obtained as a beige powder (461 mg, 92% yield).

**Synthesis procedure of 4-arm-N$_3$-PEG20k.** Synthesis of 4-arm-Ms-PEG20k: The synthesis was carried out based on a previously reported procedure[49]. 4-arm-OH-PEG20k (6.00 g, 0.300 mmol, 1.00 eq.) was

dissolved in 40 mL anhydrous DCM under $N_2$, and triethylamine (1.66 mL, 12.0 mmol, 40.0 eq.) was added. The solution was cooled to 0 °C in an ice bath, and then MsCl (0.930 mL, 12.0 mmol, 40.0 eq.) was added dropwise. The resulting mixture was removed from the ice bath and stirred for 24 h at room temperature under $N_2$. After quenching with 3 mL water, the reaction mixture was diluted with 150 mL DCM and washed with brine three times (200 mL each). The organic phase was dried over anhydrous $Na_2SO_4$ and concentrated under reduced pressure to a small volume. It was precipitated into 500 mL cold diethyl ether with vigorous stirring, filtrated and dried. The product was obtained as a colorless powder (5.26 g, 88% yield).

Synthesis of 4-arm-$N_3$-PEG20k: 4-arm-Ms-PEG20k (5.26 g, 0.263 mmol, 1.0 eq.) and $NaN_3$ (1.37 g, 21.05 mmol, 80 eq.) were dissolved in 40 mL anhydrous DMF under $N_2$. The mixture was stirred for 48 h at 60 °C under $N_2$. Then the reaction mixture was diluted with 400 mL DCM and washed with brine three times (400 mL each). The organic phase was dried over anhydrous $Na_2SO_4$ and concentrated under reduced pressure to a small volume. It was precipitated into 600 mL cold diethyl ether with vigorous stirring, filtrated, and dried. The product was obtained as a colorless powder (4.84 g, 81 % overall yield).

### Hydrogel device manufacturing and chemical composition

We use the star-shaped PEG (sPEG) abbreviation when we refer to the hydrogel formed in the click reaction of 4-arm-DBCO-PEG20k and 4-arm-$N_3$-PEG20k.

**Molding of hydrogel objects.** We prepared custom-shaped Teflon molds by using 2D computer-aided design (CAD) models that were milled with predefined depth (0.75 or 1.50 mm) into $xyz = 60 \times 30 \times 10$ mm³ Teflon blocks using a computer numerical control (CNC) machine with a 1 mm diameter milling tool. The pregel aqueous solutions were filled into the molds and polymerized. The sPEG gelation happened 15 min after the addition of precursors at room temperature, the pAAm-based gels were polymerized by 2 min UV exposure (NailStar UV lamp $4 \times 9$ W bulbs, $\lambda_{max} = 365$ nm). The pregel solutions were made freshly by mixing the stock solutions of the individual components. These stock solutions were made weekly and stored in the fridge, except the citrate buffer, HCl, NaOH, and BCP solutions that were made monthly, and the 4-arm-DBCO-PEG20k that was made daily. The exact compositions are discussed in Section "Chemical composition of the hydrogel parts". The sPEG mixture was prepared in transparent vials, by adding the 4-arm-$N_3$-PEG20k and then the urease last. The pAAm-based mixtures were prepared in brown vials, by adding the urease last. The reactants (apart from the gel-forming monomers and crosslinkers) were not covalently immobilized in the hydrogel. The gel device shapes and dimensions are shown in Fig. S1.

**Fabrication of multi-material objects.** In contrast to the main text, the following protocols are explicit about the used crosslinkers, PEGDA6k or Bis. The multi-material hydrogel objects were made step-by-step.

In the bilayer, the p(AAm-co-AA-co-PEGDA6k) layer was prepared first in a separate mold with UV irradiation. Then it was replaced into the bottom half of a double-deep mold, and the sPEG pregel was filled into the top half of the mold. After complete gelation, the bilayer was removed from the mold and superglued onto the lid of the experimental chamber filled with silicone oil.

In the mimosa bilayer, the response layer was made of two parts. First, the p(AAm-co-Bis) passive stem was photopolymerized in a separate rectangular mold. Then it was replaced into the half-deep mimosa mold, and the p(AAm-co-AA-co-Bis) leaves were photopolymerized around it. Then this whole unit was replaced into the bottom half of a double-deep mold, and the sPEG layer was prepared as in case of the simple bilayer.

In the strain gate configuration, we used one single mold but polymerized the parts sequentially. First the head of the touching patch was filled with p(AAm-co-DMAEAAm-co-Bis) and photopolymerized. Then, next to the already formed structure, we pipetted p(AAm-co-AA-co-PEGDA6k) in the neck of the touching patch and photopolymerized that as well. Then the rest was filled with the sPEG pregel, and we waited for 15 min (Fig. S2). After the sPEG gelation, the multi-material gel was removed from the mold, parafilm-covered paper clamps were superglued on the sides, and the gel was clamped on both sides in a manual stretcher by holding the paper clamps.

In the combination of the bilayer and strain-gate (Fig. 3h, i), these two units were prepared separately and then the bilayer was placed and stuck onto the already clamped strain-gate without further gluing. In general, it is key for the gel device preparation that the urease-containing layer is never photopolymerized because the radical formation destroys the urease enzyme activity. During the actuation and stretching experiments, the gels were immersed in silicone oil in a Plexi chamber with a transparent wall.

**Chemical composition of the hydrogel parts.** The sPEG front layer (Figs. 2–4) was made from 2 wt% 4-arm-DBCO-PEG20k and 2 wt% 4-arm-$N_3$-PEG20k. The p(AAm-co-AA-co-PEGDA6k) response layer (Figs. 2, 3) was made from 510 mM AAm, 100 mM AA, 6.1 mM PEG-DA6k, and 0.1 wt% LAP. The p(AAm-co-AA-co-Bis) response layer (Supplementary Movie 3) was made from 1000 mM AAm, 100 mM AA, 11 mM Bis, and 0.1 wt% LAP. These correspond to $(mon_1 + mon_2)$:cross = 100 molar ratio and 8.0 wt% total gel material content. The p(AAm-co-Bis) basis of the strain-gate (Fig. 3) was made of 1100 mM AAm, 11 mM Bis, and 0.1 wt% LAP, this composition was adjusted to have the same $(mon_1 + mon_2)$:cross = 100 molar ratio and 8.0 wt% total gel material content. The p(AAm-co-DMAEAAm-co-Bis) activator tip or patch (Figs. 2–4) was made from 5 vol% DMAEAAm and 95 vol% above p(AAm-co-Bis) mixture. In addition to the above polymerized content, the device parts also contained reactants of the embedded CRN being dissolved in the aqueous phase of the hydrogel (but not being covalently immobilized). These components are urea, urease, citrate buffer (from 0.1 M, pH = 3 stock solution), HCl or NaOH, BSA, BCP, Fmoc-EDA, and ThT. In the monolayer experiments (Fig. 2c, d), the initial concentrations were [buffer] = 1 mM, [BSA] = 0.5 g $L^{-1}$, and [BCP] = 0.075 g $L^{-1}$. In the rest, we set [buffer] = 2 mM and [BSA] = 1 g $L^{-1}$ in all gel parts. The BSA was added to promote urease stability in the solution. The indicator content was [BCP] = 0.075 g $L^{-1}$ in sPEG gels and 0.300 g $L^{-1}$ in pAAm gels because it partially decomposes during the photopolymerization. The other varied concentrations are listed in Table 1.

### Data collection and analysis in experiments

**Image recording and processing.** The color timelapse images were recorded through the transparent window of the experimental chamber by a digital camera (Panasonic Lumix G70) every 60 s. The setup was illuminated by a homogeneous white LED backlight for better image quality. The image processing and movies were made by Fiji and Microsoft PowerPoint software.

**Hydrogel stretching.** A custom-made stretcher was designed and produced in the Precision Mechanical Workshop of the University of Mainz. For assembly, the unstretched hydrogel object was clamped on both sides with parafilm-covered paper stripes (to avoid direct contact between paper and gel). It was laid flat on the stretcher, and then fixed with screwable metal vises at the parafilm-paper clamps. Then the stretcher with the gel sample was carefully placed under silicone oil. During stretching, a hand-operated screw mechanism moved the end-supports in opposite directions, keeping the center of the gel sample in the same position. Screwing in the opposite direction returned the gel to its original position.

**Instrumental analysis.** The pH of the pregel solutions was set by using a combined pH electrode (Mettler Toledo). The tensile tests were made by a Shimadzu EZ-LX universal tester using dogbone-shaped gel objects with a thickness of 1.0 mm and a middle part size of 10 mm × 3.5 mm. We stretched the dogbone samples until breaking with a speed of 200 mm min⁻¹. Each strain-stress curve was obtained from three parallel measurements. The Young's modulus ($E$, stiffness) was extracted from the slope of the linear regime of the stress-strain curves, as well as the toughness ($U_T$, deformation energy) from the integrated area under the curve. Oscillatory shear rheology was performed in an Anton-Paar MCR 702 rheometer with 8 mm (top) and 50 mm (bottom) diameter parallel plates with a gap size of 1.8 mm. The gap was filled with 120 µL sPEG pregel (containing urea, urease, and Fmoc-EDA) and surrounded by silicone oil to avoid desiccation. The kinetics of sPEG gel formation (spontaneous) and gel stiffening (initiated by a droplet of 0.1 M NaOH) were monitored by recording the storage modulus ($G'$) in time. The oscillatory shear strain ($\gamma$) was set to 1 % and the angular frequency ($\omega$) to 6.28 rad s⁻¹ based on preliminary frequency- and amplitude sweeps in the $\omega = 0.0628$–62.8 rad s⁻¹ regime at $\gamma = 0.1$ % and the $\gamma = 0.0101$–10.1 % regime at $\omega = 6.28$ rad s⁻¹. Confocal laser scanning microscopy (CLSM) was performed on Leica Stellaris 5 microscope (LasX v4.3.0.2430) with four laser lines and three HyD S detectors using plan-apochromat objectives (63×, 1.40 numerical aperture, oil immersion).

## Finite element simulations

We modeled the deformation of the strain gate upon stretching and the spatiotemporal propagation of an embedded autocatalytic reaction-diffusion (RD) front. The simulations were conducted using the structural mechanics module, transport of diluted species module, and chemistry module of COMSOL multiphysics 6.0. A hyperelastic Neo-Hookean material model was applied with a hydrogel bulk Poisson's ratio of $\nu = 0.4$ and Young's moduli $E = 18$ kPa. The symmetric horizontal stretching of the strain gate was simulated using the prescribed displacement of the two clamping patches (gray rectangles in Fig. 3a). These were moved with constant velocity in opposite directions until the gap distance in the middle ($g$) reached zero ($t_1$). The position was held for a predefined short time (until $t_2$) and then released with the same speed as stretching. We incorporated the RD phenomena using a formal autocatalytic process (1-2):

$$A \rightarrow 2A \tag{1}$$

$$\frac{\partial a}{\partial t} = k_1 a + D\Delta a, \tag{2}$$

where $a$ is the space and time-dependent concentration of the A species, $D = 10^{-9}$ m²s⁻¹ is the diffusion coefficient, and $\Delta$ is the Laplace operator. We applied no-flux boundary conditions at the outer boundary of the strain gate and an impermeable barrier at the boundary of the activator patch and the main body. The initial concentrations were $a_{patch} = 0.1$ M and $a_{body} = 0$ M. These remained unchanged before touching, i.e., at $t < t_1$ and $g > 0$. During the touching ($t_1 < t < t_2$), we applied a temporary source term for A in the touching point (TP) to simulate the contact-induced initiation of the RD front (3):

$$\dot{a}_{TP}(g) = \begin{cases} 0 \text{ molm}^{-1}\text{s}^{-1} & \text{if } g > 0 \\ 10^{-5} \text{molm}^{-1}\text{s}^{-1} & \text{if } g = 0 \end{cases} \tag{3}$$

where $\dot{a}_{TP}(g)$ is the gap-distance-dependent flux of A in the touching point. Due to this enabled local source, an autocatalytic front started to propagate from the touching point all over the main body of the strain gate. The source was disabled when the stretching was released, i.e., at $t > t_2$ and $g > 0$, but the front remained sustained because of the autocatalytic reaction.

## Data availability

The data that support the findings of this study are available from the corresponding author upon request. Supplementary Information is available. Source data are provided with this paper.

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

## Acknowledgements

This work was funded through an ERC Consolidator Grant (101001638). B.D. thanks the Alexander von Humboldt Foundation for a postdoctoral fellowship. We thank T. Gäb for initial experiments, S. Seitel for helping in the synthesis of some compounds, and A. Stenglein (Precision Mechanical Workshop, University of Mainz) for constructing the stretcher.

## Author contributions

A.W. conceived the project. B.D. designed and performed the experiments and conducted data analysis and visualization. B.D., O.S., and G.F. developed the hydrogel molding technique. O.S. and B.D. conducted the FES. G.F. synthesized the LAP and the PEGDA6k. C.L. synthesized the DBCO-COOH. A.W. and B.D. wrote the manuscript. All authors commented on the manuscript.

## Funding

## Competing interests

The authors declare no competing interests.
