## [Transparent Peer Review file · Nature Communications]

Mechano-Adaptive Meta-Gels Through Synergistic Chemical and Physical Information-Processing

Corresponding Author: Professor Andreas Walther

Version 0:

Reviewer comments:

Reviewer #1

(Remarks to the Author)

As a reviewer of the previous version of the manuscript, I can confirm that the authors have addressed most of my concerns. The manuscript is conceptually novel and will be of interest to researchers in the fields of soft materials, systems chemistry, nonlinear chemical systems, and possibly other related fields. I recommend publishing it as is.

Reviewer #2

(Remarks to the Author)

This paper presents a bioinspired design concept that unites the mechano-sensing and downstream actuation, which features local mechanical deformation perception with precise thresholds and chemical signal transmission with controlled dynamics. Free-standing soft robots were demonstrated to verify this assumption including pH-induced swelling bending and mechanical stiffening. The integration of embodied intelligence for mechano-to-chemo-to-function is interesting and inspiring, could provide a paradigm shift in design of smart robotic applications. I here recommend the publication if the following questions could be well addressed.

1. The idea of using nonlinear mechano-to-chemical response for signal processing is interesting, could the author provide more possible alternative CRN schemes with different chemical signal species (OH⁻ in the presented work) to enhance the versatility of this design strategy?
2. The mechano-chemo-mechano process in demonstrations of Figure 2 and Figure 3 required long durations (tens of mins) for signal transmission (RD process) or actuation (swelling), is it possible to achieve fast signal transmission and actuation using other chemical or mechanical response method?
3. The paper stated "Critically, shape changes during actuation are dictated by water exchange between the layers, and not by exchange to a surrounding bath. Hence, our devices differ profoundly from usual hydrogel bilayer actuators." How to experimentally evidence this claim?
4. The demonstrated actuators seem irreversible once triggered for once. Is there any possible design to enable reversible, or at least multiple actuation events, as the case for mimosa plant?

Version 1:

Reviewer comments:

Reviewer #2

(Remarks to the Author)

The author has well addressed the concerns, I recommend the publication of this paper as it is.

Reviewer #1

As a reviewer of the previous version of the manuscript, I can confirm that the authors have addressed most of my concerns. The manuscript is conceptually novel and will be of interest to researchers in the fields of soft materials, systems chemistry, nonlinear chemical systems, and possibly other related fields. I recommend publishing it as is.

We thank the reviewer for the valuable comments that helped improve our manuscript.

Reviewer #2

This paper presents a bioinspired design concept that unites the mechano-sensing and downstream actuation, which features local mechanical deformation perception with precise thresholds and chemical signal transmission with controlled dynamics. Free-standing soft robots were demonstrated to verify this assumption including pH-induced swelling bending and mechanical stiffening. The integration of embodied intelligence for mechano-to-chemo-to-function is interesting and inspiring, could provide a paradigm shift in design of smart robotic applications. I here recommend the publication if the following questions could be well addressed.

We thank the reviewer for the positive evaluation of our work.

1. The idea of using nonlinear mechano-to-chemical response for signal processing is interesting, could the author provide more possible alternative CRN schemes with different chemical signal species (OH⁻ in the presented work) to enhance the versatility of this design strategy?

This point was raised in the previous revision round in the context of ref. 36; and we had already amended the text. The following part highlights that other chemistries can be used as well. We also cited reviews on chemical reaction networks, from which a reader could select other chemistries. We do not feel that we should go into a list of CRNs that could be used, simply, because this cannot be exhaustive or appropriate.

"Such a design is rare but can be achieved using other chemistries as well. An autocatalytic front-mediated hydrogel bilayer has been constructed using thiol fronts in the front layer and reduction of disulfide crosslinks in the response layer. There the actuation mechanism is different, based on the release of the mechanical strain of the initial bilayer state.³⁶"

In general, we have good reasons to prefer the OH⁻-autocatalytic CRN. First, the availability of OH⁻-activated downstream processes in polymer chemistry and materials science, and second, the relatively fast response in the downstream process (i.e., swelling due to pH change). Using the generic pH front will be attractive for future users because it is easy to couple it with downstream processes as there are many pH responsive systems.

2. The mechano-chemo-mechano process in demonstrations of Figure 2 and Figure 3 required long durations (tens of mins) for signal transmission (RD process) or actuation (swelling), is it possible to achieve fast signal transmission and actuation using other chemical or mechanical response method?

Nontrivial life-like behaviors always arise from non-equilibrium operation, i.e., the system needs to be kept in its high-energy metastable state. Ours is a closed system, without energy or chemical input during operation, so it has a limited time before spontaneous activation. The lifetime of the dormant state and the front speed are inversely related. Also, to engineer adaptive materials with fast operation and large response, the system needs to be at the brink of stability. The lifetime of the metastable state and the response speed and magnitude can be tuned in the expense of each other, and the systems that we show in our paper represent a good compromise.

This relevant point has already been discussed in the manuscript:

"The actuation speed is limited by the front propagation that is engineered to be slow enough to provide sufficiently long-lasting dormant state, and the water transport responsible for swelling. Therefore, CRN-empowered soft robots have slower response and smaller generated forces than hard robots."

Using different chemistry or mechano-response has pros and cons. In our actuator, the response (swelling) is mediated by water transport, but as we mention in the manuscript, in ref. 36 "... the actuation mechanism is different, based on the release of the mechanical strain of the initial bilayer state.³⁶" This is another strategy, however, maybe less generic than the pH-induced swelling. We already pointed to this difference in the manuscript.

3. The paper stated "Critically, shape changes during actuation are dictated by water exchange between the layers, and not by exchange to a surrounding bath. Hence, our devices differ profoundly from usual hydrogel bilayer actuators." How to experimentally evidence this claim?

Classical textbook examples of bilayer actuators operate in aqueous environment. In the introduction of our manuscript, we referred to several such systems (refs. 10, 13, 14, 22, 26-28, etc.). In our systems presented in this work, we prepare (polymerize) the gels and place them into inert oil, without equilibration in water (see 3.2. Fabrication of Multi-Material Objects in Supplementary Information). We can clearly observe in experimental timelapse snapshots that during actuation the response layer (right) becomes swollen while the front layer (left) shrinks (Fig. 2i). A control experiment confirms that desiccation of gel objects does not occur in the silicone oil bath on the timescale of experiments (Fig. S4). We added the following sentence to the manuscript.

"The thickness of the response layer increases by 22 % while that of the front layer decreases by 22 % during the actuation (compare Fig. 2i, 0 min and 120 min)."

4. The demonstrated actuators seem irreversible once triggered for once. Is there any possible design to enable reversible, or at least multiple actuation events, as the case for mimosa plant?

As already mentioned in the text, resettability is a true challenge in such systems. We had emphasized this point also during the first round of revisions. One-time events are however also interesting when looking at mechano-adaptation as it could prevent catastrophic failure when exposed to supercritical strain.

"Our gel devices presented here respond to one-time events. Resettability and cyclic operation would be extremely challenging in such a closed, compartmentalized system. The current state of pH CRNs does not provide suitable activator-inhibitor type chemical dynamics for closed systems; a potential solution could be external refueling using vasculature or further hydrogel elements."

Resetting non-equilibrium systems is already a big challenge in solution (e.g., refueling or reproducing the initial reactants, handling waste accumulation). It is an even bigger challenge in materials systems. It can be addressed by using a **complex CRN with suitable positive and negative feedback steps**, such as in the oscillatory Belousov-Zhabotinsky (BZ) gels. However, such a mechanism (i.e., oscillatory kinetics in a closed system) is very rare, and not known yet for pH systems. One could think of the coupling of the autocatalytic urea-urease reaction with a negative feedback step (e.g., an antagonistic enzymatic reaction). Such a CRN, theoretically, could produce chemical waves that restore the chemical composition behind the propagating wave. However, in a closed system, the kinetic requirements of such dynamics (reaction rate, delay between the coupled processes) are particularly difficult to fulfil, and according to our knowledge there is no such operating system yet.

Resetting and sustained pH waves are possible in **open systems**, where the initial reactants are refueled in the gel from continuously renewing liquid reservoirs (ref. 23 in MS), but with such a solution we would lose other advantageous features of our system, e.g., the actuator efficiency would be smaller in a swollen gel state, and the integrability of the whole gel device would be weaker together with the liquid reservoir. As a future perspective, resettability in materials systems most likely requires an integrated vasculature or more complex compartmentalization.